# Epidemiological Characteristics of *Staphylococcus Aureus* in Raw Goat Milk in Shaanxi Province, China

**DOI:** 10.3390/antibiotics8030141

**Published:** 2019-09-08

**Authors:** Weidong Qian, Lanfang Shen, Xinchen Li, Ting Wang, Miao Liu, Wenjing Wang, Yuting Fu, Qiao Zeng

**Affiliations:** School of Food and Biological Engineering, Shaanxi University of Science and Technology, Xi’an 710021, China

**Keywords:** *Staphylococcus aureus*, raw goat milk, antimicrobial susceptibility, biofilm, virulence gene

## Abstract

Goat milk has been frequently implicated in staphylococcal food poisoning. The potential risk of raw goat milk contaminated by *Staphylococcus aureus* (*S. aureus*) in Shaanxi province of China is still not well documented. This study investigated the prevalence, antibiotic resistance, as well as virulence-related genes of *S. aureus* from raw goat milk samples in Shaanxi, China. A total of 68 *S. aureus* isolates were cultured from 289 raw goat milk. Most of the isolates were resistant to penicillin and oxacillin, although 41.18%, 33.82%, and 29.41% of the isolates expressed resistance to piperacillin, trimethoprim-sulfamethoxazole, and ciprofloxacin, respectively. Our data demonstrated that 91.18% of the isolates produced biofilm, of which 54.41% isolates belonged to high-biofilm producers. In addition, genotypic analysis of biofilm related genes (*fnbA, clfB, fnbB, cna*) revealed that 91.18% of the isolates harbored at least one of the genes, in which the most prevalent genes were *fnbA* (66. 17%), *clfB* (48.53%), and *fnbB* (26.47%). 94.8% of the isolates contained at least one toxin-related gene, of which *seb* (76.47%), *tsst* (36.76%), and *sea* (23.53%) genes were the more frequently detected. Further analysis revealed a positive association between *fnbA*, *clfB*, *fnbB*, *seb*, *tsst,* and *sea* genes and certain antibiotic resistance. The results indicated that raw goat milk samples contaminated by *S. aureus* can be a potential risk to public health.

## 1. Introduction

*Staphylococcus aureus* (*S. aureus*) is a Gram-positive pathogen that can be isolated from a wide range of food, animal, human, and medical environments. Recently, microbial food poisoning accounted for 53.7% of the food poisoning incidents in China, of which *S. aureus* is one of the most commonly identified foodborne pathogens that causes a wide range of clinical infections [1]. *S. aureus* has been described to contaminate various foods such as chicken, fish, dairy products, sushi, and sashimi [2,3,4,5,6], and thus constitutes a risk for consumer health. In addition, in the dairy industry, *S. aureus* can be introduced at almost each step of the production process, and therefore staphylococcal food poisoning caused by *S. aureus* has also been highlighted by the dairy industry [7].

*S. aureus* is a highly versatile bacterium adapting to hostile environments, which was inseparably associated with its ability to form structures known as biofilms [8,9]. A biofilm is defined as an assemblage of microbial cells that irreversibly attaches to a surface and enclosed in a matrix of hydrated extracellular polymeric substances [10]. Biofilm development of *S. aureus* consists of a three-stage process of deeply wired genetic developmental process triggered by stress signals, including (i) attachment of cells to surface, (ii) maturation of the biofilm, and (iii) detachment/dispersal [11,12]. The initial attachment to abiotic or biotic surfaces for biofilm formation involves numerous surface-anchored proteins such as biofilm-associated protein (*bap*), collagen-binding proteins (*cna*), clumping factors A and B (*clfA*, *clfB*), as well as fibronectin-binding proteins A and B (*fnbA*, *fnbB*), which are collectively termed as microbial surface components recognizing adhesive matrix molecules (MSCRAMM) [13,14,15,16], providing a critical step to establish infections. Accumulation/maturation of biofilm is mediated by the polysaccharide intercellular adhesin encoded by the *icaABCD* gene cluster [13]. *S aureus* cells growing in a biofilm are physiologically distinct from planktonic cells, which improve their survival and growth during food processing, thereby establishing a competitive advantage for *S. aureus* as an etiological agent of foodborne infectious diseases [15]. Biofilms are very difficult to eliminate once they have been established. Furthermore, biofilms physically limit access of antibacterial agents to bacteria. Therefore, bacteria within biofilms are up to 10–1000 times more resistant to antimicrobials than if they were planktonic cells without biofilms [17].

Staphylococcal food poisoning is directly associated with toxins and invasive enzymes produced by *S. aureus* such as staphylococcal enterotoxin (SE), toxic shock syndrome toxin-1 (TSST-1), and Panton-Valentine leucocidyn (PVL) [18]. Twenty-three types of SEs (Sea-See and Seg-Sev) have been described [19]. SE is divided into five classical serological types (Sea, Seb, Sec, Sed and See), and more than 90% of *S. aureus*-associated food poisoning outbreaks were associated with these classic SEs [20]. Additionally, TSST-1 toxin can lead to toxic shock syndrome by reducing host immune response, and PVL can cause tissue necrosis by destroying host leukocytes [21]. Other SEs (Seg-Seu) were also detected and have been implicated in staphylococcal food poisoning [22].

Foodborne outbreaks of *S. aureus* intoxications have been documented to be associated with consumption of contaminated milk [23,24]. Goat milk is the third largest source of dairy production in the world, accounting for 2.07% of the total milk. The top goat milk producing countries in the world are as follows: India, Bangladesh, Pakistan, and Sudan. China is rich in goat milk, in which Shaanxi province is one of the main sources of goat milk in the Chinese market [25]. With the gradual popularization of goat milk, raw goat milk might be a potential source of staphylococcal food poisoning caused by *S. aureus*. However, only a few studies have been performed on *S. aureus* prevalence and contamination levels in raw goat milk in China [26]. Meanwhile, there are a paucity of data regarding SE gene distribution and biofilm formation ability of *S. aureus* isolates from raw goat milk samples in Shaanxi province, China. This study aims to investigate *S. aureus* contamination in raw goat milk in Shaanxi province, China. For this purpose, we analyzed the prevalence, contamination levels, antibiotic susceptibility profiles, virulence genes, biofilm formation, and biofilm-related genes of the *S. aureus* isolates obtained from Shaanxi province, China.

## 2. Results

### 2.1. Isolation and Identification of S. aureus

Of the 289 raw milk samples, 68 (23.53%, 68/289) were confirmed to be *S. aureus* in this study. *S. aureus* counts ranged from 2.6 × 10^2^ to 3.3 × 10^4^ CFU/mL, with a mean value of 5.6 × 10^3^ CFU/mL, in which 25 isolates were lower than 10^3^ CFU/mL and 43 isolates were more than 10^3^ CFU/mL, suggesting the existence of factors associated with the *S. aureus* in raw goat milk produced in the region.

### 2.2. Antibiotic Resistance of S. Aureus Isolated from Raw Goat Milk

The resistance profile of isolates to the tested antimicrobial agents is presented in Table 1. Resistance to penicillin was the most common (79.41%, 54/68), followed by oxacillin (60.29%, 41/68), piperacillin (41.18%, 28/68), trimethoprim-sulfamethoxazole (33.82%, 23/68), ciprofloxacin (29.41%, 20/68), gentamicin (27.94%, 19/68), clindamycin (20.59%, 14/68), cefazolin (19.12%, 13/68), and linezolid (14.71%, 10/68). Specifically, 66 isolates were resistant to at least one antibiotic (97.06%, 66/68) and 36 isolates (52.94%, 36/68) were resistant to three or more antibiotics (MDR). All the *S. aureus* isolates were susceptible to vancomycin.

### 2.3. Biofilm Formation and Observation

The results showed 62 isolates (91.18%, 62/68) have the ability to produce biofilms, of which 37 isolates (54.41%, 37/68) were high-biofilm producers, 21 isolates (30.88%, 21/68) of medium-biofilm producers, 4 isolates (5.88%, 4/68) of low-biofilm producers, and 6 isolates of non-biofilm producers.

For visualization of biofilm formation in *S. aureus*, biofilms of representative *S. aureus* isolates were further investigated by scanning electron microscope (SEM) and confocal laser scanning microscopy (CLSM) (Figure 1). The biofilm formation ability is presented in Figure 1. Biofilms grown on the coverslip conglomerated in thick, heterogeneous and multiple layers with columnar clusters were observed in SA-134, which was high-biofilm producer (Figure 1A,E,I,M). Biofilms clustered into a honeycomb structure were observed in SA-135, which was medium-biofilm producer (Figure 1B,F,J,N). By contrast, the SA-136 isolate of low biofilm-producer exhibited lower degree of staining and showed more dispersed (Figure 1C,G,K,O). Almost all cells of the SA-137 isolate showed a single-cell distribution, which was non-biofilm producer (Figure 1D,H,L,P).

### 2.4. Adhesion and Biofilm-Related Genes

The presence of *S. aureus* adhesion and biofilm associated genes was detected by PCR (Figure 2, Appendix A). The *fnbA* gene was observed in 66.18% (45/68) of the isolates, followed by the *clfB* gene (48.53%, 33/68), the *fnbB* gene (26.47%, 18/68), the *cna* gene (13.24%, 9/68), while *bap*, *icaA*, and *icaD* genes were not detected in the isolates. Overall, 59 isolates (86.76%, 59/68) in 68 strains contained at least one adhesion and biofilm-related genes.

### 2.5. Virulence-Related Genes

The prevalence of virulence-related genes was shown in Figure 3. Of the virulence genes investigated, the three most frequently detected virulence genes were *seb* (50.00%, 34/68), *tsst* (36.76%, 25/68), and *sea* (23.52%, 16/68). The prevalence of the other virulence genes was as follows: *seq* (14.70%, 10/68), *sec* (10.29%, 7/68) and *seu* (7.35%, 5/68), and *ser* (1.47%, 1/68). However, genes of *sed*, *see*, *seg*, *seh*, *sei*, *sej*, *sek*, *sel*, *sem*, *sen,* and *seo* were not detected in isolates studied here.

### 2.6. The Association between Biofilm Formation and Biofilm-Related Genes

In the present study, there were a significant association between the presence of *fnbA* (*p* < 0.001), *clfB* (*p* < 0.05), along with *fnbB* (*p* < 0.05) genes and biofilm formation in *S. aureus* isolates. Nonetheless the association between *cna* gene and biofilm formation were not observed (*p* > 0.05). In addition, the *fnbA*, *clfB*, *fnbB,* and *cna* genes were simultaneously detected in 5 (7.4%, 5/68) isolates, of which 4 isolates were biofilm producers. Five (7.4%, 5/68) tested isolates were negative for *fnbA*, *clfB*, *fnbB,* and *cna*, and only one isolates formed biofilms.

### 2.7. The Relationship between Antibiotic Resistance and Biofilm-Related Genes

As was shown in Figure 4A, the presence of *fnbA* gene was significantly correlated with resistance to penicillin (*p* < 001) and gentamicin (*p* < 0.01), the presence of *fnbB* gene was significantly associated with resistance to ciprofloxacin (*p* < 0.001) and gentamicin (*p* < 0.001) (Figure 4B). The presence of *clfB* gene was significantly associated with resistance to trimethoprim-sulfamethoxazole (*p* < 0.01) and penicillin (*p* < 0.001) in *S. aureus* isolates (Figure 4C), while the association was not observed for *cna* gene (*p* > 0.05). 

### 2.8. The Relationship between Antibiotic Resistance and Virulence Genes

The present study revealed a significant association between the presence of *sea* gene and resistance to gentamicin (*p* < 0.001) and oxacillin (*p* < 0.001) (Figure 5A). The presence of *seb* gene was strongly related with resistance to oxacillin (*p* < 0.01) and penicillin (*p* < 0.001) (Figure 5B). And the presence of *tsst* gene was obviously associated with resistance to gentamicin (*p* < 0.05) and penicillin (*p* < 001) in *S. aureus* isolates (Figure 5C).

## 3. Discussion

The contamination of dairy products by *S. aureus* isolates, especially those expressing an MDR phenotype and being capable of producing biofilms and toxins such as enterotoxin, TSST-1, and PVL, poses a serious public health threat to humans. This has been manifested by food-borne poisoning outbreaks resulting from the contamination of *S. aureus*, including one of the largest staphylococcal food poisoning outbreaks on record involving 13,420 infected individuals in Japan. Similarly, food-borne infections caused by contaminated dairy foods by *S. aureus* were also frequently described in China [27]. Here, this study investigated the prevalence, the antibiotic resistance phenotype, the ability to form biofilms, along with the presence of biofilm-associated genes and virulence genes of *S. aureus* isolates from farms in Shaanxi province, China. Acquisition of the prevalence and characteristics of isolates will contribute to prevent the contamination of raw goat milk by *S. aureus* via effective intervention and protect the end consumer.

In this study, 68 out of 385 raw goat milk samples (17.6%, 68/385) were positive with *S. aureus*. The prevalence of *S. aureus* is significantly lower than that in the previous report that showed the contamination rate of *S. aureus* in pork industry was 26% (130/501) in China [28]. Similarly, it was also obviously less than that in other reports which demonstrated that *S. aureus* was detected in 53.5% (153/286) of the bulk tank milk samples in Italy [29], and 46% (47/104) of the bulk tank milk samples in the United States [30]. In addition, a higher contamination rate of *S. aureus* was 76.9% (60/78) in bulk tan milk samples in Italy [31]. However, Xing et al. [26] found that the prevalence of *S. aureus* in raw goat milk of healthy goats in Shaanxi province was 1.5% (1/67) of in final products and 7.5% (60/781) of in goat milk powder processing plant environments in 2012–2013, respectively, which is significantly less than the result obtained here. This increasing prevalence of *S. aureus* observed here might be attributed to the fact that misuse of antibiotics and several other factors has contributed to the increasing emergence of multidrug-resistant *S. aureus*. Thus, huge development should allow for controlling *S. aureus* contamination in raw goat milk.

The rapid emergence of antimicrobial resistance has threatened to render our current antibiotic therapeutics arsenal useless and poses a great challenge in treatment of serious bacterial infections. In the current study, the antimicrobial susceptibility test results showed that the 97.06% (66/68) of the *S. aureus* isolates from raw goat samples had different degrees of resistance to 10 antimicrobials (Table 1). 52.94% (36/68) of isolates were multi-resistant to three or more antimicrobial agents. The results obtained here were lower than 87% of the isolates from pasteurized milk samples in China [32], 88% of the isolates from cattle milk, and 87% of the isolates from sheep milk samples in Jordan [33]. Besides, resistance to penicillin (79.41%), oxacillin (60.29%), and piperacillin (41.18%) were the most frequently observed, which are the commonly used antibiotics in veterinary drug therapy. In the current study, 79.41% of the isolates from raw goat milk [34] showed resistance to penicillin G; the penicillin resistance levels were higher compared to the two previous reports in China (62%), where samples were from large Chinese dairy herds [35], and in South Africa (71.6%) where *S. aureus* isolates were derived from animal carcasses and milk samples from the abattoirs and dairy farms [36]. Resistance to oxacillin were significantly higher than the samples from mastitis-affected cows in India (0%) [37]. In addition, 33.82% of the isolates studied here showed resistance to trimethoprim-sulfamethoxazole. By contrast, Unal et al. [38] reported that all isolates from raw goat milk were sensitive to trimethoprim-sulfamethoxazole and cefoxitin. The relatively high rates of resistance and MDR isolates observed in this study may be due to the frequent use of penicillin-based antimicrobials in goat farms.

Recently, diverse virulence gene profiles were reported from different food categories worldwide, such as fishery products in Galicia. Virulence gene patterns investigated here showed that 79.4% (54/68) of the isolates were positive with SE genes, which is more than that in the reports of 50% of *S. aureus* isolates [39]. Among the enterotoxin SE genes, the most prevalent gene was *seb* (76.47%), which was higher than the results from a previous report (11.69%) [40]. Several studies described that none of isolates from bovine, goat milk and dairy products harbored *seb* gene [41,42,43]. Detection rate of *sec* gene was 10.3% in our report, which is lower in goat milk by others reported. Maslankova et al. [44] also reported that *sec* gene was the most frequently detected (24.1%) in *S. aureus* from sheep milk and dairy products. The different prevalence rates observed in SE gene might be explained by fact that these isolates originated from geographically distinct locations [45]. Of note, the *tsst* gene (36.76%) was detected here, which was significantly higher than that in the report of 2.2% of *S. aureus* isolates [39]. Even though most the thermal-treating processes for food processing are strong enough to inactivate any vegetative *S. aureus* cells present, but their enterotoxins have displayed their activity even after these processes [46]. Therefore, more emphasis should be focused on the microbiological examination of milk and dairy foods.

This study also investigated the formation of biofilms, adhesion genes, and biofilm-related genes in all isolates studied. In this study, 91.18% (62/68) of *S. aureus* isolates had ability to form biofilms, which was in congruence with studies from other countries [47]. Among adhesion genes tested, *fnbA* (66.18%), *clfB* (48.53%), and *fnbB* (26.47%) genes were observed with a high detection rate. Prevalence of the *fnbA* gene were consistent with the results of Zmantar et al. (76.1%) [48]. The *fnbA* and *fnbB* are *S. aureus* cell surface-bound proteins, bind to both fibronectin and fibrinogen, which enable *S. aureus* to attach to host cells of another organism. In addition, *clfB* gene encoding binding protein that facilitates *S. aureus* to establish nasal colonization was also present in 48.53% of the isolates in our study. The report [49] was higher (93%) than our results. Therefore, our results and other published reports suggest that *fnbA*, *clfB,* and *fnbB* exhibit an important role in *S. aureus* biofilm formation. The percentage of *cna* gene in this study was 13.24%, which is lower than previous reports by Pereyra et al. and Klein et al. [50,51]. Ciftci et al. [52] found that different frequencies of *icaA* and *icaD* have been reported, but *icaA* and *icaD* genes are not detected in our study. Our findings showed that the prevalence of adhesion and biofilm-related genes varied widely among isolates. 

An interesting finding of this study was that the presence of *fnbA* (*p* < 0.001), *clfB* (*p* < 0.05), and *fnbB* (*p* < 0.05) genes showed a significant association with biofilm formation. The MSCRAMM family includes fibronectin binding proteins (*fnbA* and *fnbB*), collagen binding protein (*cna*), and human fibrinogen binding protein aggregation factors (*clfA* and *clfB*). Several other surface molecules such as teichoic acid may also be important for direct adsorption to the surface of the object [53]. However, they do not play a very important role in the infection of medical devices implanted in the body. This means that *fnbA*, *fnbB,* and *clfB* genes play a vital role in the formation of biofilms.

In this study, a significant correlation was observed between the presence of *fnbA* gene and resistance to gentamicin (*p* < 0.001) and penicillin (*p* < 0.01). Similarly, there was a positive correlation between the presence of *fnbB* gene and resistance to ciprofloxacin (*p* < 0.001) and penicillin (*p* < 0.001). The results also showed that the presence of *clfB* gene was correlated positively with trimethoprim (*p* < 0.01), respectively. In addition, the presence of *sea* gene was significantly associated with resistance to gentamicin (*p* < 0.001) and oxacillin (*p* < 0.001), while the presence of *seb* gene was significantly associated with oxacillin (*p* < 0.01) and penicillin resistance (*p* < 0.001). Noteworthy, we found a significant association between the presence of *tsst* gene and resistance of gentamicin (*p* < 0.05) and penicillin (*p* < 0.001) in *S. aureus* isolates, respectively. Overall, there is a positive association between certain antibiotic resistance and biofilm related gene as well as virulence gene in *S. aureus* isolates.

## 4. Materials and Methods 

### 4.1. Isolation and Identification of Strains

A total of 289 raw goat milk samples were collected from three goat farms during 2016 to 2017 in Shaanxi province, China. Raw milk samples were collected from goats. One milk sample from each goat was collected. The *S. aureus* contamination was detected using the most probable number (MPN) method in raw goat milk samples according to National Food Safety Standards of China document GB 4789.10-2016. Briefly, 25 ml milk sample was taken and transferred into 225 mL 10% (w/v) saline solution, and further culture was streaked onto Baird–Parker Agar supplemented with 5% egg yolk and blood agar with sterile defibrinated sheep blood (Land Bridge, Beijing, China), respectively, then incubated at 37 °C for 48 h, the total number of colonies on each plate was counted. Finally, all isolates were subjected to the detection of 16S rRNA. All confirmed *S. aureus* isolates were stored in tryptic soy broth (TSB) with glycerol at −80 °C. No more than two isolates of each sample were chosen for subsequent studies.

### 4.2. Antimicrobial Susceptibility Testing

The Kirby–Bauer disk diffusion method was applied to test the antibiotic susceptibility of all isolates. Diameter were interpreted by the guidelines of Clinical and Laboratory Standards Institute [CLSI] (2017). All isolates were assessed for antimicrobial susceptibility to 10 antibiotics (Oxoid, United Kingdom). The antimicrobial agents included gentamicin (CN, 10 μg), ciprofloxacin (CIP, 5 μg), penicillin (PCN, 10 μg), cefazolin (KZ, 30 μg), clindamycin (DA, 2 μg), vancomycin (VA, 30 μg), oxacillin (OX, 1 μg), piperacillin (PRL, 110 μg), trimethoprim-sulfamethoxazole (SXT, 25 μg), and linezolid (LZD, 30 μg). *S. aureus* ATCC 25923 was used as the control strain for antimicrobial susceptibility testing.

### 4.3. Biofilm Formation and Classification

The microtiter plate assay for investigating biofilm formation is a means which permits the observation of bacterial adherence to an abiotic surface. Slightly, after overnight incubation in TSB, 200 μL of cell suspension (approximately 1 × 10^8^ CFU/mL) was diverted into each well and incubated at 37 °C for further 24 h. After three washes with PBS (0.01 mol/L, pH 7.4) and a 20-min fixation step with 100% methanol, dyed with 0.4% (*w/v*) crystal violet (CV) for 15 min and washed with PBS (0.01 mol/L, pH 7.4). Bound crystal violet was then dissolved with 33% acetic acid for 30 min. The biofilm was measured at 580 nm of OD in a microplate reader (Thermo Fisher Scientific, Finland). The biofilm assays were performed in triplicate. According to the critical ODc value, the biofilm is divided into the following four categories: OD ≤ ODc of non-biofilm producers, ODc < OD ≤ 2 ODc of low-biofilm producers, 2 ODc < OD ≤ 4 ODc of medium-biofilm producers, and OD > 4 ODc of high-biofilm producers.

### 4.4. Scanning Electron Microscope and Confocal Laser Scanning Microscopy Analysis of Biofilm

For visualization of biofilms, biofilms were further observed by a scanning electron microscope (SEM, phenom pro, Thermo Fisher Scientific, Netherlands). After overnight growth of *S. aureus*, 1 mL of cell suspension (approximately 1 × 10^8^ CFU/mL) was transferred into a 24-well plate containing coated glass coverslips, cultured at 37 °C for another 48 h, and fixed with 2.5% glutaraldehyde-PBS solution for 2 h at 4 °C. The coated glass coverslips were then dehydrated in a series of washes with 30, 50, 70, 90% ethanol for 10 min each, followed by 15 min rinses in 100% ethanol. Air-dried samples were immediately sputter-coated with platinum and SEM analysis was performed.

Confocal laser scanning microscopy (CLSM, LSM800, Carl Zeiss AG, Germany) was performed to observe more clearly the biofilm of *S. aureus*. Bacteria biofilms were prepared using the same methods as described above. The isolates were removed from the medium and gently rinsed with 0.9% to remove medium and unattached cells. The cells were then stained with carboxyfluorescein diacetate succinimidyl ester (CFDA-SE, Beyotime, China) for 10–15 min. Cells were visualized and photographed using CLSM. The excitation/emission wavelength was 488/542 nm.

### 4.5. DNA Extraction and Gene Detection of PCR

Isolates were incubated overnight at 37 °C in TSB medium. The genomic DNA was extracted using the Genomic Extraction Kit (Beijing Trans Gen Biotechnology Co., Ltd.), and the quantity and quality of DNA was determined by a NanoDrop-2000 spectrophotometer (Thermo Fisher Scientific, NH, USA). The adhesion (*fnbA*, *fnbB*, *clfB*, *cna*), biofilm-associated (*bap*, *icaA*, *icaD*) and virulence-related genes (*sea*, *seb*, *sec*, *sed*, *see*, *seg*, *seh*, *sei*, *sej*, *sek*, *sel*, *sem*, *sen*, *seo*, *seq*, *ser*, *seu*, *tsst*) were detected by PCR, respectively. The primers were supplied by Sangon Biotech Co., Ltd. (Shanghai, China) (Appendix A).

### 4.6. Data Analysis and Statistical Analysis

All experiments were examined independently, statistical analysis was performed using SPSS Statistics 22 software, graph Pad Prism 5 was used to make graphics. The data were compared using independent sample test and ANOVA, *p* < 0.05 indicates that the difference was statistically significant.

## 5. Conclusions

In summary, our study provides an epidemiological investigation of the prevalence of *S. aureus* in raw goat milk in Shaanxi province, China. This study indicates a high prevalence of *S. aureus* with virulence genes and biofilm forming ability that might lead to the persistence of *S. aureus* in goat farms contributing to serious infections and subsequent food poisoning. To the best of our knowledge, our results firstly found a significantly positive correlation between biofilm related gene as well as virulence gene and antibiotics resistance. The results provide potential alternative monitoring measures by gene detection to eliminate and reduce the contamination of *S. aureus* in raw goat milk.

## Figures and Tables

**Figure 1 antibiotics-08-00141-f001:**
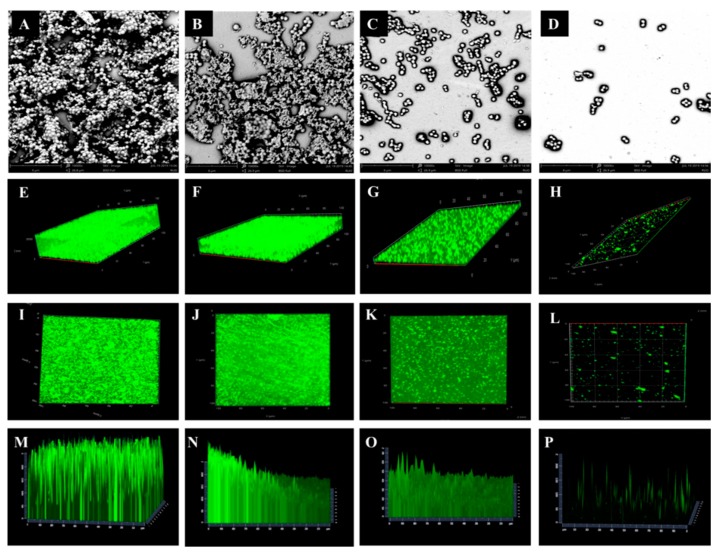
Scanning electron microscope (SEM) (**A**–**D**) and confocal laser scanning microscopy (CLSM) (**E**–**P**) images of biofilm of representative *S. aureus* isolates. A, D, I, M: high-biofilm producer (SA-134); B, F, J, N: medium-biofilm producer (SA-135); C, J, K, O: low-biofilm producer (SA-136); D, H, L, P: non-biofilm producer (SA-137).

**Figure 2 antibiotics-08-00141-f002:**
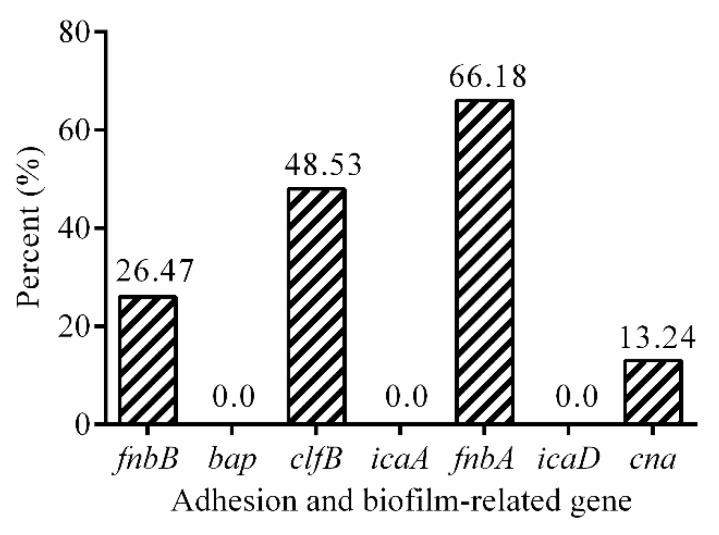
The prevalence of adhesion and biofilm-related genes in 68 isolates.

**Figure 3 antibiotics-08-00141-f003:**
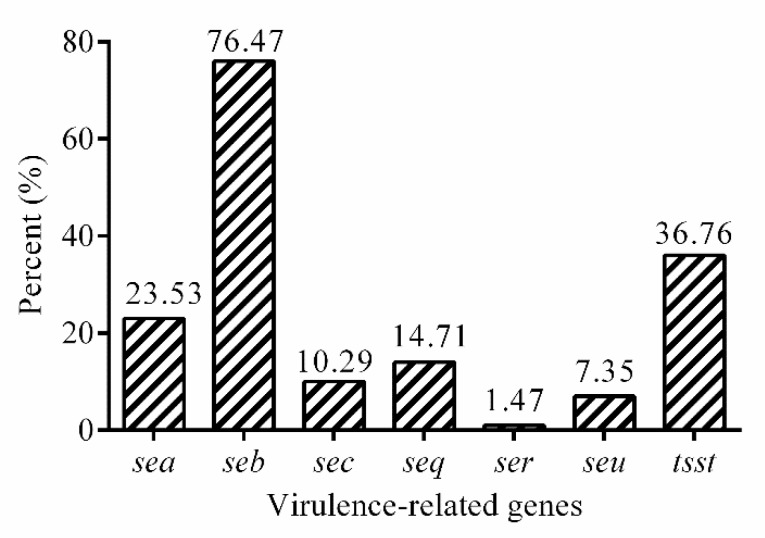
The prevalence of virulence genes in 68 isolates.

**Figure 4 antibiotics-08-00141-f004:**
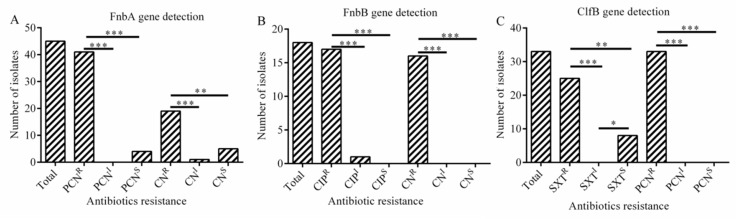
The relationship between adhesion as well as biofilm-related genes and antibiotic resistance (**A**–**C**). (**A**) Correlation between *fnbA* gene and antibiotic resistance. (**B**) Correlation between *fnbB* gene and antibiotic resistance; (**C**) Correlation between *clfB* gene and antibiotic resistance. Statistics were achieved by independent sample test and ANOVA (*** *p* < 0.001, ** *p* < 0.01, * *p* < 0.05). PCN^R^: resistance to penicillin; PCN^I^: intermediary to penicillin; PCN^S^: sensitive to penicillin; CN^R^: resistance to gentamicin; CN^I^: intermediary to gentamicin; CN^S^: sensitive to gentamicin; CIP^R^: resistance to ciprofloxacin; CIP^I^: intermediary to ciprofloxacin; CIP^S^: sensitive to ciprofloxacin; SXT^R^: resistance to trimethoprim-sulfamethoxazole; SXT^I^: intermediary to trimethoprim; SXT^S^: sensitive to trimethoprim.

**Figure 5 antibiotics-08-00141-f005:**
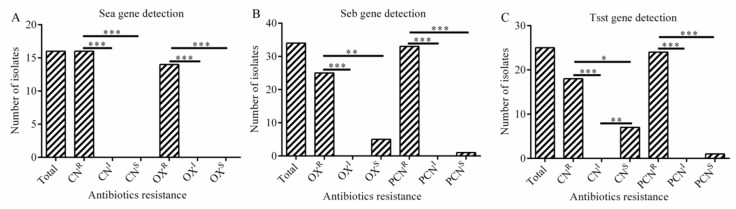
The association between virulence gene and antibiotic resistance (**A**–**C**). (**A**) Correlation between *sea* gene and antibiotic resistance; (**B**) Correlation between *seb* gene and antibiotic resistance; (**C**) Correlation between *tsst* gene and antibiotic resistance. Statistics were achieved by independent sample test and ANOVA (*** *p* < 0.001, ** *p* < 0.01, * *p* < 0.05). CN^R^: resistance to gentamicin; CN^I^: intermediary to gentamicin; CN^S^: sensitive to gentamicin; OX^R^: resistance to oxacillin; OX^I^: intermediary to oxacillin; OX^S^: sensitive to oxacillin. PCN^R^: resistance to penicillin; PCN^I^: intermediary to penicillin; PNC^S^: sensitive to penicillin.

**Table 1 antibiotics-08-00141-t001:** Antimicrobial susceptibility test of the 68 *S. aureus* isolates obtained from raw goat milk samples.

Antibiotic	Content (µg)	Diameter of Inhibition Zone (mm)			Susceptibility and Resistance (%)		
		S	I	R	%	%	%
Trimethoprim-sulfamethoxazole	25	≥21	-	≤20	45 (66.18)	0 (0)	23 (33.82)
Linezolid	30	≥16	11–15	≤10	57 (83.82)	1 (1.47)	10 (14.71)
Gentamicin	10	≥15	13–14	≤12	48 (70.59)	1 (1.47)	19 (27.94)
Ciprofloxacin	5	≥21	16–20	≤15	43 (63.24)	5 (7.35)	20 (29.41)
Cefazolin	30	≥18	15–17	≤14	50 (73.53)	5 (7.35)	13 (19.12)
Clindamycin	2	≥21	15–20	≤14	40 (58.82)	14 (20.59)	14 (20.59)
Vancomycin	30	≥15	-	≤16	68 (100.00)	0 (0)	0 (0)
Penicillin	10	≥29	-	≤28	14 (20.59)	0 (0)	54 (79.41)
Oxacillin	1	≥18	-	≤17	27 (39.71)	0 (0)	41 (60.29)
Piperacillin	110	≥18	-	≤17	40 (58.82)	0 (0)	28 (41.18)

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
