# Peer review of "Epidemiological Characteristics of Staphylococcus Aureus in Raw Goat Milk in Shaanxi Province, China"

_antibiotics, 2019, doi:10.3390/antibiotics8030141_

Round 1

Reviewer 1 Report

The authors of this manuscript, Epidemiological characteristics of Staphylococcus aureusin raw goat milk ....  , emphasize the danger of SA in raw milk products. They mention SA a versatile microorganism. That may be true in some aspects, however for causing a foodborne intoxication it is necessary that SA grows in a product rich in proterins (that is raw goat milk), preferably in absence of high numbers of competitive microorganisms (which is not often the case in raw milk). High numbers of SA may expected when milk is drawn from animals suffering from mastitis.

The toxin cannot be formed in milk stored at low temperatures (10-12C).

The authors isolated 68 SA isolates from ? samples raw goat milk, but did not mention their (log) count per ml milk. I missed also a section of material and methods.

Moreover, the authors mixed medical and foodborne information (literature). I think it is better to concentrate on foodborne!

The main work has been done on antibiotic resistance, virulence gene and biofilm formation. However, the reader remains unclear about the level of contamination of raw milk. Usually counts of lower than 500 SA/ml are accepted, sometimes even more. 

The manuscript can be improved if it is placed in a broader context. 

Reviewer 2 Report

Epidemiological characteristics of Staphylococcusaureus in raw goat milk in Shaanxi Province, China

Microbial contamination in food is a major concern in our daily life. In this manuscript, the author studied the prevalence, antibiotic resistance and its correlation with associated gene expression, which may provide important clues on S. aureusdetection and intervention in milk production chain, as well as draw our attention on microbial contamination, antibiotic resistance and food safety.

The manuscript is well-written in a brief format and the data are clearly presented and convincing. I recommend its publication in Antibiotics.

Here, I have some relatively minor comments.

Line 99, the group of non-biofilm producer is SA-137 isolate? The same isolate as low biofilm producer. I think it may be a mistake by authors. For figure 4 and 5, it is really confused. Why not the author use bar to indicate the significance between groups? The a,b,c here is somehow cryptic to show which group with which. The gene detection is evaluated by PCR. Could it be possible to show some gel band? Is there any difference of brightness of bands? For example, the biofilm related gene, could the brightness of PCR bands somehow correlate to the enrichment of aureusbetween isolates and the degree of biofilm production?

Author Response

请参阅附件

Reviewer 3 Report

In the manuscript entitled "Epidemiological characteristics of Staphylococcus
aureus in raw goat milk in Shaanxi Province, China", Qian and colleagues describe the isolation of a total of 68 s. aureus isolates from raw goat milk from Shaanxi, and the characterization of their resistance profiles to common antimicrobials. Furthermore, the authors also provide information concerning the ability of isolates to form biofilms, as well as the presence/ absence of genes known to play a role in adhesion/biofilm formation, and virulence. The autohrs also analysed the relationships between antibiotic resistance and presence of genes related to biofilm, as well as between antibiotic resistance and virulence genes. Overall, the work is well organized, the English language can be improved, and some parts require a revision to improve the quality of the work. In my opinion, the work would better fit into a Food/Food Microbiology journal, as the main focus of the work is milk contamination by S. aureus. A major issue is related with the "Results" part. In my opinion, an introductory section mentioning the sampling, methods of detection of S. aureus and the levels of contamination of samples (e.g. CFU/mL) is necessary prior the antibiotic resistance section. A series of minor issues follows:

Minor issues

line 19: "... biofilm related genes...." the genes analysed should be listed after "genes"

line 22: "were the more frequently detected"

line 23: "Further analysis revealed a positive"

line 29: " pathogen that can be isolated from a wide "

line 39: " irreversibly attach to a surface"

lines 47-48: correct references or delete names of authors.

line 52: " for S. aureus as an etiological agent of foodborne infectious diseases"

line 53: "established. Furthermore, biofilms physically limits"

line 58: "Panton-Valentine leucocidyn"

line 60: "and more than 90%..."

line 67: give a reference

line 71: "raw goat milk in China." give a reference.

Table 1 : instead of "NO. (%)2 use "%"

Table 1 legend: "Antimicrobial susceptibility test of the 68 S. aureus isolates obtained from raw goat milk samples"

lines 107-108: the sentence "To further investigate the phenotypic differences, the presence of S. aureus adhesion and biofilm associated genes was evaluated by PCR (Figure. 2). "  is inadequate, as the authors did not try to correlate phenotypes with the presence of genes, only the percentages of the total isolates were considered. Rephrase or analyse according to the biofilm phenotype.

Figures 2 and 3: genes names should be in italics.

Round 2

Reviewer 1 Report

Well, the questions are answered and where asked some extra explanation was added.

I wonder why a MPN method was used for counting Stapylococcus, and not the BP-RPF enumeration method, which is common for raw milk samples. Te results of the investigsated samples shows that the goat milk is of a rather good quality (for S. aureus). The extension of the paper to biofilms - and antibiotic resistance genes is less important, I guess. Cleaning and Disinfection in a good way should prevent the growth of biofilms in food industry and medical settings. I am not sure that the characterisation of staphylococci in biofilms will prevent this. Of sourse, it is interesting to know all possibilities of the isolated strains, and it is worthwhile to publish it. However, it is extension of our knowledge, but not for the prevention of ...

The last remark could be used as an additional point for the conclusion.   

Reviewer 3 Report

The Authors have made a thorough effort to review the manuscript, adding important pieces of information that were lacking, namely strian isolation and identification. This reviewer has no further criticisms to the manuscript.